# Integration of Distributed Energy Resources and EV Fast-Charging Infrastructure in High-Speed Railway Systems

Miad Ahmadi [1], Hamed Jafari Kaleybar [2,*], Morris Brenna [2], Francesco Castelli-Dezza [1] and Maria Stefania Carmeli [1]

1   Department of Mechanical, Politecnico di Milano, 20156 Milan, Italy; miad.ahmadi@polimi.it (M.A.); francesco.castellidezza@polimi.it (F.C.-D.); stefania.carmeli@polimi.it (M.S.C.)
2   Department of Energy, Politecnico di Milano, 20156 Milan, Italy; morris.brenna@polimi.it
*   Correspondence: hamed.jafari@polimi.it

**Abstract:** Low carbon emission transportation is attracting global attention where electric railway power systems (ERPS) and electric vehicles (EVs) act as a load. Besides the main utility grid, renewable energy sources (RES) including photovoltaic (PV) panels and wind turbines are implemented to supply the loads fully or partially. In this paper, a novel smart DC catenary system is proposed in which renewable sources, storage systems, and DC fast-charging stations are connected to the overhead DC catenary line of the high-speed railway power system. The generated power from renewable sources and consumed power by charging stations are processed by their dedicated DC-DC power electronics converters. Furthermore, a storage system is used as a backup system not only for the case of blackouts but also because of the intermittent nature of renewable energy sources to supply the loads continuously. The paper presents an optimal power control for various parts and a power management system (PMS) that manages the power flow from wind-PV-storage system to EV-ERPS system. The proposed system has been investigated using a real Italian Rome-Florence 3 kV high-speed line as a case study with real data of ERPS load. The EV fast-charging station power demand, wind speed, solar irradiance, and temperature were recorded for 24 h in order to provide us with realistic output data. The simulation results obtained by MATLAB/Simulink are presented to validate the effectiveness of the proposed system.

**Keywords:** DC railway microgrid; renewable energy sources; EV fast-charging infrastructures; energy storage systems; DC hub

## 1. Introduction

Nowadays, countries are in great need of green and low carbon emission transportation; hence, transportation electrification has been given worldwide consideration [1]. The evolution to electrified transportation has been reflected by the emergence of electric vehicles (EV), hybrid electric vehicles (HEV), electric railway systems, etc. [2], and new technologies such as Digital Twin (DT) are currently being introduced into this market for better implementation [3]. Electric railway power systems (ERPS) have been introduced to the market from the beginning of the 20th century and they have been developed over time; additionally, they are operating in the form of high-speed trains, regional trains, subways, tramways, electric buses, etc. [4]. However, ERPS have been increasing over time and they have become one of the large-scale consumers in the utility grid, which puts a great burden on the main grid. Moreover, the usage of electric vehicles has shown growth that initiated the further installment of charging stations that result in more load burden on the utility grid [5]. On the other hand, the world is facing fossil fuel diminution that highlights the necessity of applying environmentally friendly power generation [6]. By this means, renewable energy sources (RES) have been progressively utilized for clean power generation, in which wind and solar energies are the favorable choices.

During the last years, the importance and feasibility of improving ERPS through implementation of smart grid concept has been significantly received attention by experts [7]. Moreover, integrating DERs in the ERPS would contribute to partial independence from the main utility grid suppliers. Despite the fact that uncertain and stochastic modes of power generation for renewable energy sources (RESs) are the major problematic issues, there exists a relatively high trend to integrate them into railway infrastructures. In References [8–10] the possible scenarios of integrating PV sources to the AC railway feeder are investigated. A smart metro station as a DC microgrid integrated with PV and energy storage systems (ESSs) is discussed in [11]. An integration scheme to reduce the subscribed power and to remove the related overrun by the RES has been evaluated in [12] for the metro system. These researches are discussed in AC or low-voltage DC ERPSs considering one type of renewable resource. The authors of [13] have presented a medium-voltage dc (MVDC) multiterminal ERPS with the capability of transferring energy in 24 KV DC hub. However, this system has been discussed as a paradigm or suggested model and the power flow control has not been mentioned. In the proposed model of this study, wind and PV generators are integrated into a MVDC railway catenary system as a smart DC hub which can be complementary on a day-to-day and season-to-season basis. Nevertheless, due to the uncertainty of wind and PV power generation, a storage system is interlinked with the system to warrant the grid-like power in case of power shortage [14]. On the other hand, and to the best of the authors' knowledge, integrating EV fast-charging infrastructure to the railway DC hub has not been addressed technically so far. In the proposed smart DC catenary system, a charging station is integrated into the overhead catenary system, in which charging station power demand is proceed through the catenary system and then bidirectional DC-DC power electronics converter. In order to validate the effectiveness of the system, real input data for 24 h are used. Solar radiation and temperature are obtained from real measurements at an optimum location close to the Rome—Florence 3 kV dc overhead catenary system. Likewise, an optimum location for installing a wind turbine is chosen where the daily wind speed is measured. Considering the trains' power demand, a realistic profile of daily power demand at one substation is adopted for this study. Ultimately, the data that are used for the charging station's power demand is obtained from a charging station near traction power system substation (TPSS) in Florence where the area was filled with industrial companies. Within this study, PV and the storage unit are integrated into the DC catenary system through a bidirectional DC-DC converter because of two major grounds. First and foremost, the reliability and redundancy of the PV-storage system will significantly increase, as the PV-storage DC voltage will be raised to 3 kV dc through a high-frequency transformer (HFT) of the dual active bridge rather than connecting many PV or storage cells in series to reach to medium voltage values [15]. In the second place, galvanic isolation will be provided between the PV-storage system and the rest of the system. The main contributions of this paper can be defined as follows: the possibility of the integration of large-scale renewable energy sources (wind turbines and PV panels), charging infrastructures, and storage units with novel configuration into the DC catenary system of the ERPS is evaluated. Additionally, increment of the system capacity and compensating voltage drop of the $3\ kV_{dc}$ ERPS is validated by transforming the DC catenary function into a DC hub. Results also present that for increasing the system capacity and decreasing the voltage drop especially during peak hours (due to increasing of traffic), the integration of distributed system proposed in this paper is a favorable option rather than changing or updating the system structure that is quite costly and unacceptable for newly established structures.

The rest of the paper is formed as follows: Section 2 explains the principle of the studied system and shows the architecture of the proposed system. In Section 3, a detailed description is given about various parts of the system and their control scheme. A power management system that is implemented for this study is defined in Section 4. The final outcome and simulation results are presented in Section 5 and ultimate Section 6 is the conclusion of the paper.

## 2. Principles and Configuration of DC High-Speed Railway Systems

Railway infrastructures electrification have encountered a substantial adjustment and evolution procedures in last decades. A rapid increment in social population and additional demand for high-capacity and high-speed transportation systems in line with historical and geographical potentials of the countries have led to various configurations of ERPSs across the globe. In the first place, ERPSs were based on the DC supplying system with multi-pulse rectifications systems in TPSS. Afterwards, by developing of power system and electrical motors technologies, the AC ERPSs are emerged and became favored. The foremost supplying systems adapted for high-speed railway systems are 1.5/3 kV DC, 15 kV–16.67 Hz, and 25 kV–50/60 Hz. In the last century, Italy as one of the precursor countries developed the high-speed ERPSs infrastructures. They are manufactured by establishing the line between most popular cities using ETR 200 trains in 3 kV DC supply system. A real dedicated high-speed ERPS was developed in 2010 with 923 km long of operation [16]. Then, the operation with ETR 500 trains at 2 × 25 kV AC, 50 Hz system was constructed in 2012. During the last years, ETR 1000 trains have also been developed, with operation speed up to 360 km/h. Nowadays, 11,921 km of high-speed ERPS track is electrified in 3 kV DC and 1296 km are in 2 × 25 kV AC lines [16]. Given that the majority of high-speed lines are supplied by 3 kV DC, the proposed system has been investigated according to a real Italian Rome-Florence 3 kV high-speed line as a case study.

The required energy for the DC ERPSs is commonly fed through the high-voltage AC grid in primary side and converted into DC in the secondary side of TPSS. In the 3 kV DC ERPS the interval of TPSSs is between 15–40 km subject to the line data and design powers range. In the Italian line, the AC/DC conversion stage is obtained via a twelve-pulse diode rectifier bridge. In this configuration, the positive port is connected to the overhead contact wire, and the negative port is connected to the rails. In the TPSS the primary-side, AC voltage is transformed into the DC voltage of 2710 V to supply the contact wire. Every conventional TPSS contains two groups with a modular configuration and a power rating of 3.6/5.4 MW. In order to highlight the bidirectional power flow between grid and ERPS, the proposed TPSS in the paper is based on bidirectional converter.

## 3. Modeling and Integration of RES, Energy Storage System, and EV Fast-Charging Station into DC Catenary System

The proposed architecture of the smart DC catenary system is shown in Figure 1. It consists of two renewable energy sources (PV panels and wind turbines) and a storage unit that acts as a backup. The utility grid is the main supplier in the proposed electric railway power system while the EV fast-charging station and the power train are the consumers of the power. In the proposed model, PV arrays and the storage system are interlinked with their dedicated DC-DC power electronics converters that are boost and buck-boost converters, respectively.

However, a dual active bridge (DAB) converter is responsible for bi-directional power flow between the overhead DC catenary system and the common point of the PV-storage system. EV fast-charging infrastructure is connected to the DC catenary system through a DAB converter which provides the system with bi-directional power flow that is grid to vehicle (G2V) and vehicle to grid (V2G) where the latter one is considered for future developments. In the following, a detailed model of each part and its control structure is evaluated.

### 3.1. Grid Connected Converter

Figure 2 demonstrates the architecture of the AC-DC power electronics converter and its control structure. The grid-connected converter has the main role of power exchange between AC and DC sides. In the proposed smart DC catenary system of electric railway power system, bidirectional IGBT converter is applied, and the control construction is based on the voltage-oriented control (VOC). The considered AC-DC converter is connected to the main grid through resistor and inductance that act as an AC side filter.

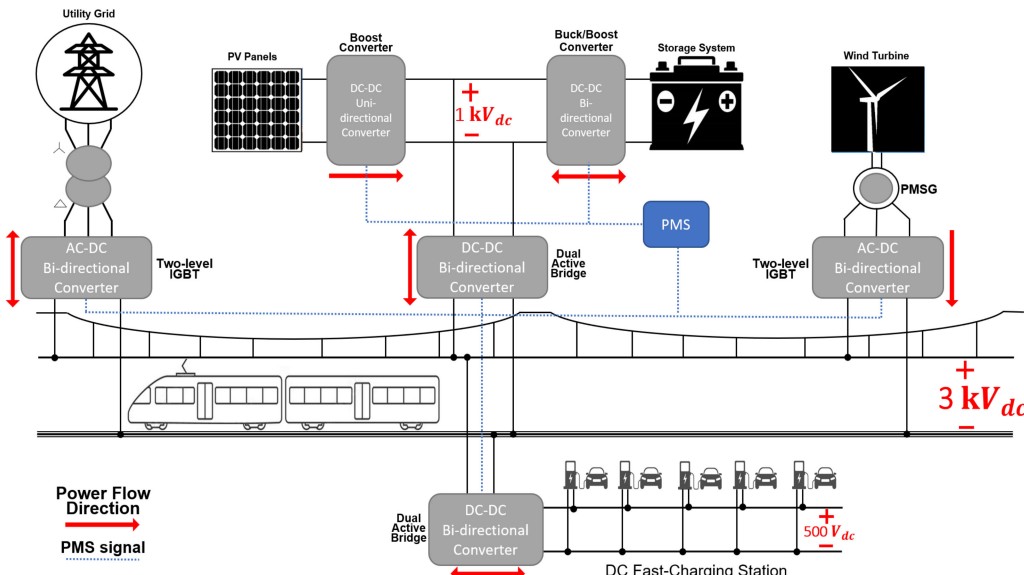

**Figure 1.** Smart DC catenary system architecture.

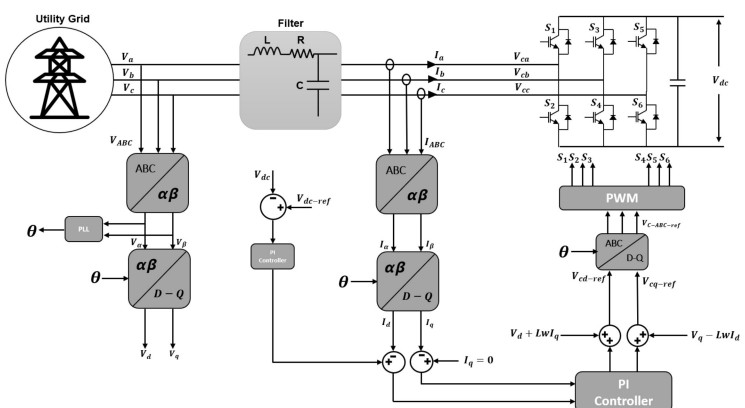

**Figure 2.** AC-DC grid connected converter with its control structure.

Utilizing α-β frame, a voltage controller is used for maintaining the DC voltage of the overhead catenary line. Additionally, a PI current regulator is used which converts voltage error signals to the currents ($i_d$, $i_q$). Therefore, for modeling the controller, ABC coordinates are transformed to *d-q* coordinates. The controller design follows the following steps accordingly [17]:

$$\begin{bmatrix} V_a \\ V_b \\ V_c \end{bmatrix} = L\frac{d}{dt}\begin{bmatrix} i_a \\ i_b \\ i_c \end{bmatrix} + R\begin{bmatrix} i_a \\ i_b \\ i_c \end{bmatrix} - \begin{bmatrix} V_{ca} \\ V_{cb} \\ V_{cc} \end{bmatrix} \tag{1}$$

$$L\frac{d}{dt}\begin{bmatrix} i_d \\ i_q \end{bmatrix} = \begin{bmatrix} -R & +\omega L \\ -\omega L & -R \end{bmatrix}\begin{bmatrix} i_d \\ i_q \end{bmatrix} + \begin{bmatrix} V_d \\ V_q \end{bmatrix} - \begin{bmatrix} V_{cd} \\ V_{cq} \end{bmatrix} \tag{2}$$

$$\begin{cases} V_{cd} = V_d - R\,i_d + \omega L i_q - L\frac{di_d}{dt} \\ V_{cq} = V_q - R\,i_q + \omega L i_d - L\frac{di_q}{dt} \end{cases} \tag{3}$$

$$If\ R \ll 1 \rightarrow \begin{cases} V_{cd} = V_d + \omega L i_q - L\frac{di_d}{dt} \\ V_{cq} = V_q + \omega L i_d - L\frac{di_q}{dt} \end{cases} \tag{4}$$

where ($i_d$, $i_q$), ($V_d$, $V_q$), and ($V_{cd}$, $V_{cq}$) are obtained from ($i_a$, $i_b$, $i_c$), ($V_a$, $V_b$, $V_c$), and ($V_{ca}$, $V_{cb}$, $V_{cc}$) respectively.

### 3.2. PV

The photovoltaic system consists of PV arrays and a DC-DC boost converter for maximum power point tracking (MPPT) that is shown in Figure 3.

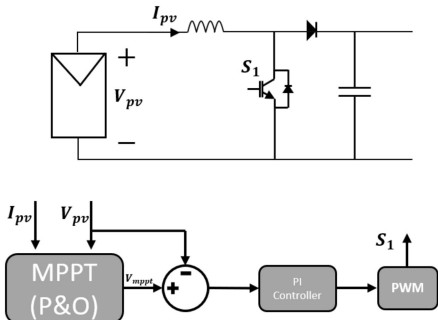

**Figure 3.** PV system with its control structure.

In boost converter connected to PV arrays, there are two operating modes based on switch $S_1$, when it is OFF and when it is ON. Hence, the output voltage of the boost converter is given by:

$$V_{dc} = \frac{V_{pv}}{1 - D} \tag{5}$$

where $V_{dc}$ is the output voltage of the boost converter, $V_{pv}$ is the output voltage of the *PV* arrays, and $D$ is the duty cycle. In Figure 3, when the switch $S_1$ is ON, the current will flow to the switch $S_1$, and the current level of the inductor increases. When switch $S_1$ is OFF, the current stored in the inductor will pass through the diode. In this study, for extracting the maximum power of PV arrays, the actual output voltage of the PV panels is compared to $V_{mppt}$ in order to find the error. This error is fed into the PI controller, and the output of the PI controller will give the required duty ratio for the PWM of the boost converter. This controller will provide a better MPP tracking performance since it has a closed-loop control. The boost converter control equation is presented in Figure 4.

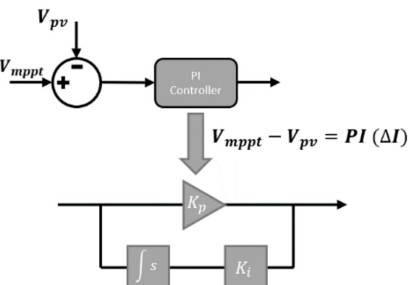

**Figure 4.** PI controller of the boost converter.

In order to reach a specific level of output power and voltage, depending on the PV cells model, a certain number of cells are connected in series and parallel. In the following, the mathematical model of PV arrays and DC-DC boost converter with its MPPT control scheme will be explained.

### 3.2.1. Mathematical Modeling of PV System

As PV cells are formed by compounding p and n type semi-conductors, the cells' essential quality is similar to diode. Hence, one-diode equivalent circuit shown in Figure 5a is the most frequent utilized model for evaluating the PV array. Generally, the one-diode equivalent circuit has five parameters and since $R_{sh}$ is large in value, in some studies, the model is further simplified to the four parameters model shown in Figure 5b [18,19].

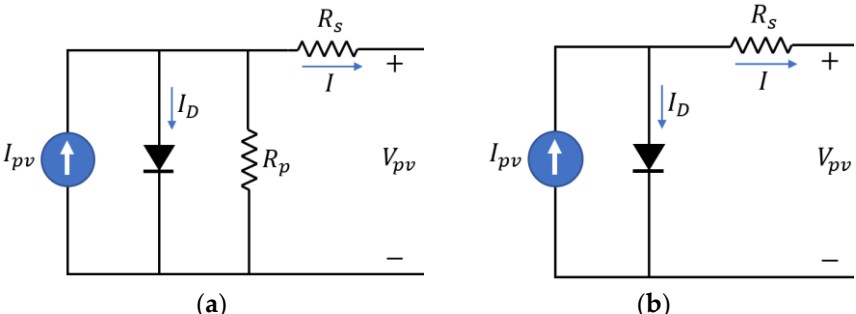

**Figure 5.** Equivalent circuit of PV arrays; (**a**) five parameters; (**b**) four parameters.

According to the semiconductor theorem and Kirchhoff's current and voltage law (KCL, KVL) of the PV circuit model in Figure 5a, the mathematical equation is [20]:

$$I = I_{pv} + I_o \left( exp\left( \frac{qV_{pv}}{akT} \right) - 1 \right) \tag{6}$$

where $I_{pv}$ is the *PV* generated current, $I_o$ is the diode saturation current, $q$ is electron charge that equal to $1.60217646 \times 10^{-19}$ C, k is the Boltzmann constant equal to $1.3806503 \times 10^{-21}$ J/k, T is the temperature between p-n junction in kelvin, and a is the diode ideality constant. Nevertheless, due to losses of real PV arrays, PV characteristics cannot be determined based on Equation (6); hence, Equation (7) is applied which considers the losses.

$$I = I_{pv} + I_o \left( exp\left( \frac{V_{pv} + IR_s}{V_t\, a} \right) - 1 \right) - \frac{V_{pv} + IR_s}{R_p} \tag{7}$$

where $V_t$ is the thermal voltage of PV arrays that consist of parallel and series number of PV cells ($N_s$) as follows:

$$V_t = \frac{N_s kT}{q} \tag{8}$$

3.2.2. Maximum Power Point Tracking Control

Due to the intermittent nature of solar radiation and environment temperature, the output power of PV panels might change. Therefore, the MPPT algorithm is applied to warrant the maximal power extraction [21]. In literature, there are various MPPT algorithms including Perturb and Observe (P&O), incremental current, artificial intelligence (AI), etc. The chosen algorithm in this paper is P&O, which is common due to its simpleness and practicability.

The function of the P&O principle and its algorithm is shown in Figure 6. As can be seen, in the P&O, the PV output voltage varies with the trivial change of irradiance that bring changes to the output power of the PV system which is shown with $\Delta P$ and it is explained as follow:

- If $\Delta P > 0$, it shows that we are getting close to MPP, so any incrementation in the same direction will shake the operating point unto MPP.
- If $\Delta P < 0$, it represents that the operating point pulls away from MPP, hence, the direction of the operating point should be reversed.

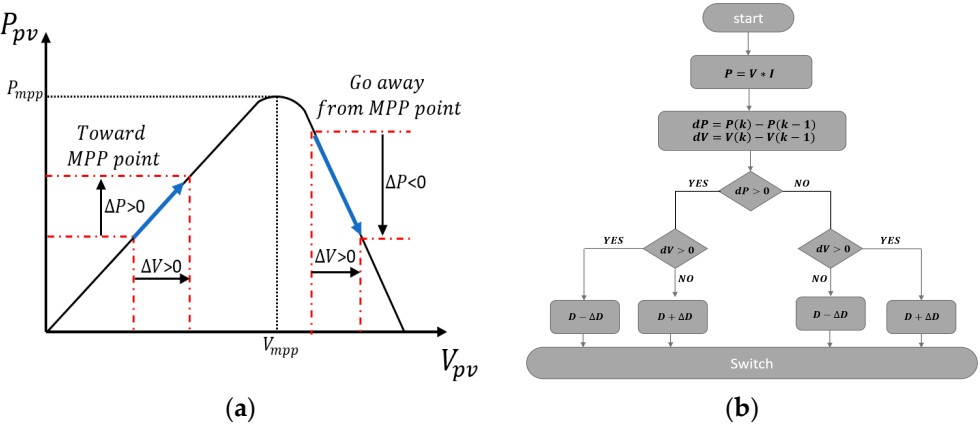

**Figure 6.** MPPT; (**a**) working principle; (**b**) algorithm.

### 3.3. Energy Storage System

The storage unit is interlinked with the system through a bi-directional DC-DC buck-boost power electronics converter which enables the unit with two ways of power flow. The existence of the storage unit is essential due to the uncertainty of renewable energy generators and as a backup in case of a blackout. The mathematical model of the storage system is founded on two substantial factors which are terminal voltage and state of the charge given by [22].

$$V_{bat} = V_o + R_{bat}I_{bat} - \frac{kQ_{bat}}{Q_{bat} + \int I_{bat}dt} + A_{bat}exp\left(B_{bat}\int I_{bat}dt\right) \tag{9}$$

$$SOC = 100\left(1 + \frac{\int I_{bat}dt}{Q_{bat}}\right) \tag{10}$$

where $R_{bat}$ is the battery resistance ($\Omega$), $V_o$ is the output voltage of battery when it is open source (V), k is the polarization voltage (V), $Q_{bat}$ is the battery capacity (Ah), $A_{bat}$ is the exponential voltage (V), and $B_{bat}$ is the exponential capacity ($Ah^{-1}$).

The storage unit configuration and its control scheme are represented in Figure 7. The operating mode of the storage system (charging/discharging) depends on the input current; if $I_{bat} < 0$, the battery will charge will $I_{bat} < 0$ will discharge the battery. In this controlling scheme, a two proportional-integral (PI) controller is implemented. The PI voltage regulator is applied for retaining the DC link voltage and its input is the error between measured voltage and the reference voltage of DC-link. The PI current regulator's input is the error between charging/discharging current reference and measured current [23]. However, the storage unit has a limitation with its SOC and most of the time, Equation (11) should be met.

$$20\% \leq SOC \leq 80\% \tag{11}$$

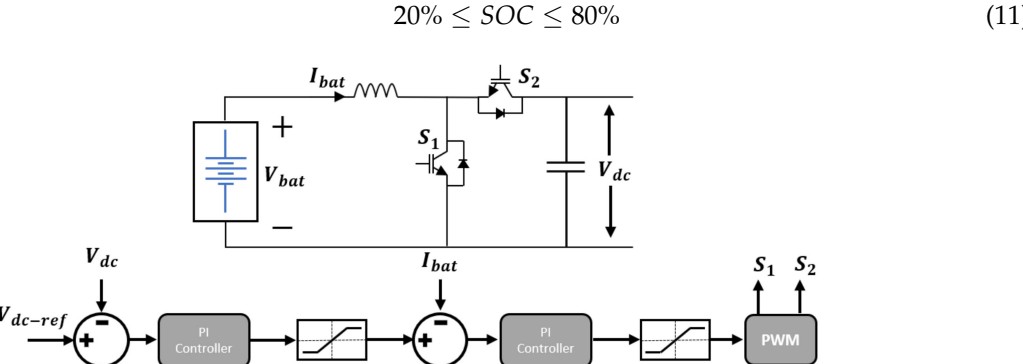

**Figure 7.** Storage system with its control structure.

In Figure 8, the block diagram of the storage unit controller is shown. The objective of the controller is to charge/discharge the storage unit according to the system requirements. Therefore, the actual value of the DC voltage ($V_{dc}$) is compared with the reference voltage and then the error is given to the PI controller. The PI controller generates the required current (charge/discharge current) from the storage unit that is called the reference current. This required current is then limited by the rate limiter in order to control the charge/discharge rate. Finally, the reference current is compared with the actual current of the battery ($I_{bat}$) and the error is given to PI controller which generates the duty ratio for switches $S_1$ and $S_2$.

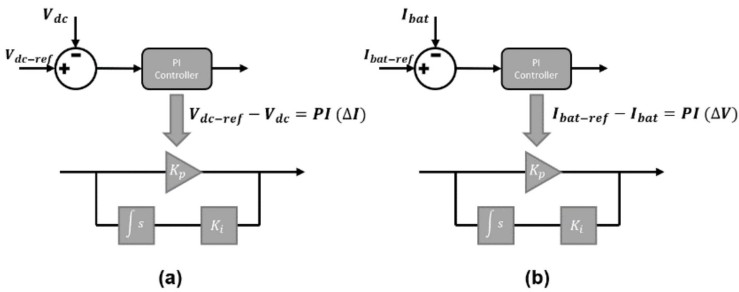

**Figure 8.** PI controller of the buck-boost converter for (**a**) current (**b**) voltage.

The buck-boost converter is modeled depending on the duty cycle D of the switches. According to the switches $S_1$ and $S_2$ the converter operation can be divided in two modes:

- When $S_1$ is ON and $S_2$ is OFF

$$\frac{di_{bat}}{dt} = \frac{V_{bat}}{L} \tag{12}$$

- When $S_1$ is OFF and $S_2$ is ON

$$\frac{di_{bat}}{dt} = \frac{V_{bat} - V_{dc}}{L} \tag{13}$$

Therefore, the converter model can be expressed by following:

$$\frac{di_{bat}}{dt} = \frac{V_{bat} - V_{dc}}{L} + \frac{V_{dc}}{L}D \tag{14}$$

### 3.4. EV Fast-Charging Station (DAB Converter)

In this study, DC fast charging station is interlinked with the smart DC overhead catenary line of the electric railway through a DAB converter that is capable of bidirectional power flow [24]. The configuration of such a system and its control structure is presented in Figure 9.

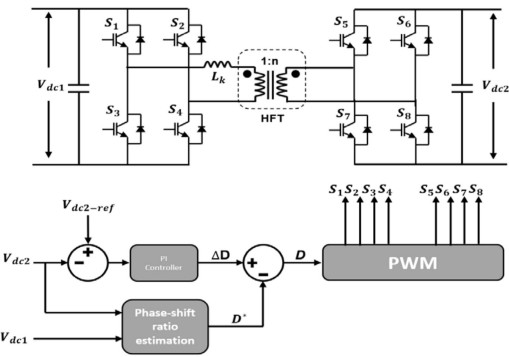

**Figure 9.** DC fast charging infrastructure (DAB converter) and its control scheme.

A dual active bridge converter is composed of two single-phase full bridges and a high-frequency transformer (HFT). This bidirectional converter transfers the energy from the DC catenary line to the EV's battery package (and vice versa in the case of V2G) by the means of phase shifting. By altering the rate of phase-shift between primary and secondary bridge, the DC-DC converter controls the power flow, and this control function is expressed as Equation (15):

$$P = \frac{V_{dc1} V_{dc2} \varphi (1 - \varphi)}{n \pi L_k \omega} \tag{15}$$

where $V_{dc1}$ and $V_{dc2}$ are the primary and secondary DC voltages respectively, $n$ is the ratio of primary and secondary windings of the HFT, $\omega$ is the angular frequency, $L_k$ is the leakage inductance referred to the primary side of the HFT, and $\varphi$ is the phase-shift value [25,26]. The equations for calculation of phase shift and leakage inductance are defined as follows:

$$\varphi = \frac{\pi}{2} \left( 1 - \sqrt{1 - \frac{8 f_s L_k P_{out}}{n V_{dc1} V_{dc2}}} \right) \tag{16}$$

$$L_k = \frac{V_{dc1} V_{dc2} d (1 - d)}{2 f_s n P} \tag{17}$$

In Equations (16) and (17), $f_s$ is the switching frequency and $d$ is the duty cycle percentage. DAB converter has an interesting topology due to its merits. It provides the system with galvanic isolation that is obligatory by many standards. Furthermore, it inherits the zero-voltage switching (ZVS) that reduces the losses and increases efficiency. Regarding the switching techniques of DAB, there are several phase-shifting techniques including single-phase shift, dual-phase shift, triple-phase shift, extended phase shift, and hybrid phase shift [24,27]. In this scrutiny, a single phase-shifting angle is selected because of its simplicity and feasibility.

### 3.5. Wind Turbine System

The wind generator is one of the main renewable energy sources that converts the wind kinetic energy into electrical energy through its wings and then a set of gear-box, generator, and power electronics converters. In this scrutiny, a permanent magnet synchronous generator (PMSG) is implemented that is interlinked with the DC catenary system through an AC-DC power converter. A simplified mathematical equation between wind speed and mechanical output power is expressed by [28]:

$$P_m = \frac{1}{2} C_p \rho \pi R^2 \mathcal{V}_{wind}^3 \tag{18}$$

where $\rho$ is the air density ($\rho$ = 1.225 kg/m$^3$), R is the turbine rotor radius (m), $\mathcal{V}_{wind}$ is the wind speed (m/s), and $C_p$ is the power coefficient that is defined as the variable of $\beta$ and $\lambda$ which are blade pitch angle and tip speed ratio, given by [29,30]:

$$C_p(\lambda, \beta) = C_1 \left( \frac{C_2}{\lambda_i} - C_3 \beta - C_4 \right) e^{\frac{C_5}{\lambda_i}} + C_6 \lambda \tag{19}$$

$$\frac{1}{\lambda_i} = \frac{1}{\lambda + 0.08\beta} - \frac{0.035}{\beta^2 + 1} \tag{20}$$

The $C_1$–$C_6$ are constant, in that their values are based on the wind turbine model and manufacturer [31]. As before mentioned, the wind turbine is connected to the DC-link through a power converter. In case of low-power generation, a diode rectifier + DC-DC

converter is used, whereas, in high power purposes, an active PWM rectifier is utilized. The dynamic relationship of permanent synchronous generator is as follows [32,33]:

$$\begin{cases} V_{sd} = -L_q p \omega_r i_{sq} + L_d \frac{di_{sd}}{dt} \\ V_{sq} = p \omega_r \varphi_v + L_d p \omega_r i_{sd} + L_q \frac{di_{sq}}{dt} \end{cases} \tag{21}$$

where $(i_{sd}, i_{sq})$ and $(V_{sd}, V_{sq})$ are the D-Q coordinates of the ABC coordinates of stator current and voltage $(i_{sa}, i_{sb}, i_{sc})$ and $(V_{sa}, V_{sb}, V_{sc})$ respectively. $p$ is the poles number of PMSG, $\varphi_v$ is the magnetic flux linkage, and $\omega_r$ is the rotor angular speed. Hence, the wind turbine generator and its control structure are presented in Figure 10.

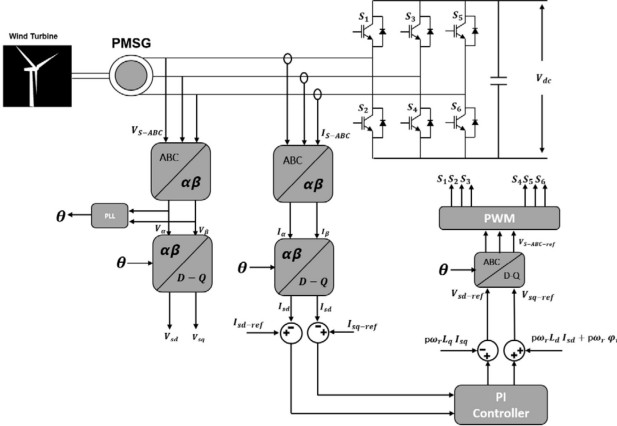

**Figure 10.** Wind turbine generator with its control scheme.

## 4. Proposed Power Management System

The proposed system operates under different working conditions and power management system (PMS) is applied to manage the power flow in order to ensure loads receive adequate power. PMS is a centralized system that communicates with all decentralized control units. In this study, PMS is supervised to work under two main configuration (grid connected and standalone) and four various operating modes that are feeding mode (FM), emergency mode (EM), peak shaving mode (PSM), and recovery mode (RM). PMS is implemented as a state machine and lookup table in the simulation. Table 1 summarizes the applied PMS operating modes.

**Table 1.** Power management system operational modes.

| Grid | ESS | $P_{renewables}$ | Working Mode | | |
|---|---|---|---|---|---|
| Supply | disconnected | Ppv = 0, Pwind = 0 | FM-1 | | |
| Supply | disconnected | Ppv = 0, Pwind > 0 | FM-2 | | |
| Supply | disconnected | Ppv > 0, Pwind = 0 | FM-3 | FM | |
| Supply | disconnected | Ppv > 0, Pwind > 0 | FM-4 | | |
| Supply | charging | Ppv > 0, Pwind > 0 | FM-5 | | Grid connected |
| Supply | discharging | Ppv = 0, Pwind > 0 | PSM-1 | PSM | |
| Supply | discharging/disconnected | Ppv > 0, Pwind > 0 | PSM-2 | | |
| Absorb | charging/disconnected | Ppv + Pwind > Pev + $P_{TPSS}$ | RM | | |
| Disconnected | charging | Ppv + Pwind > $P_{TPSS}$ + Pev | EM-1 | | |
| Disconnected | disconnected | Ppv+ Pwind = $P_{TPSS}$ + Pev | EM-2 | EM | Standalone |
| Disconnected | discharging | Ppv+ Pwind < $P_{TPSS}$ + Pev | EM-3 | | |

## 5. Simulation Results

The effectiveness of the proposed smart DC catenary system is carried out in MAT-LAB/Simulink environment for realistic daily profiles of inputs. In this paper, 24-h real profiles of electric railway power demand at a specific substation, wind speed, solar radiation, temperature, and EV charging station power demand are applied. All these measurements are performed based on real data. Figures 11–13 reveal the input data for above-mentioned parameters. As can be seen from Figure 11, solar radiation varies from $0$ W/m$^2$ at nights to about 1200 W/m$^2$ in the middle of the day with some trivial shading between 12:00 a.m. and 3:30 p.m. Daily profile of temperature is also variable between 12 °C and 36 °C. Likewise, the minimum and maximum values of the wind speed profile are 3.7 m/s and 10 m/s respectively. Considering the EV charging station, it is installed in the suburb area with many companies where the EVs get charged during working hours and the peak hours take place in the morning when the electric vehicles arrive. The daily power demand profile of the charging station shown in Figure 12 indicates that peak hour of charging station occurs from 7:00 a.m. to 11:00 a.m.

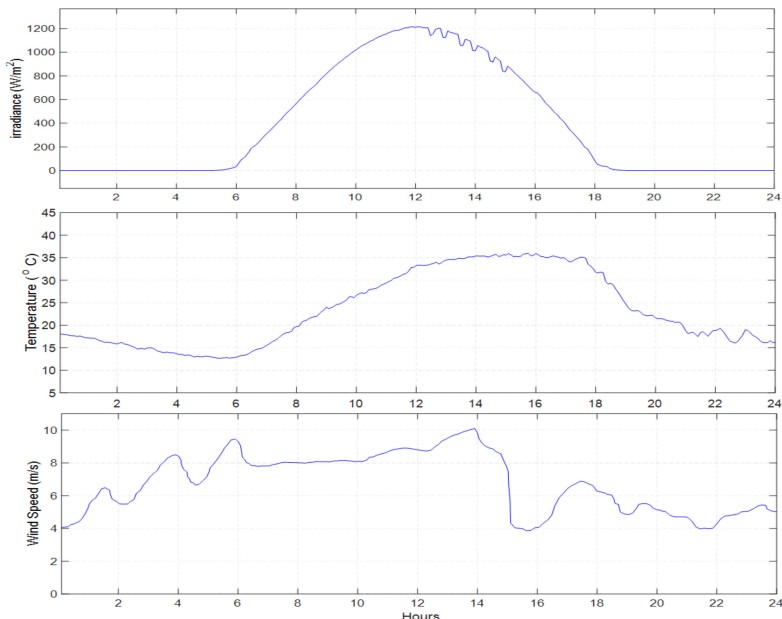

**Figure 11.** Input data for solar irradiance, temperature, and wind speed.

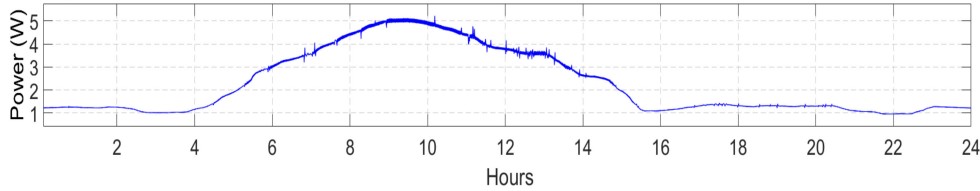

**Figure 12.** Fast charging daily power demand.

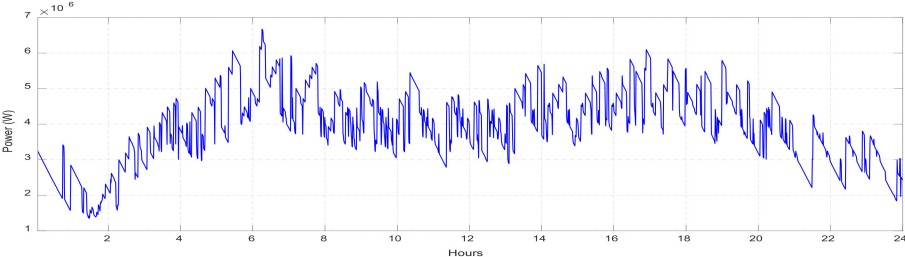

**Figure 13.** Trains daily power demand at one traction power system substation (TPSS).

In order to simulate the exact behavior of ERPS, generating/demanding powers at one $2 \times 3.6$ MW TPSS are considered. Figure 13 illustrates the daily power profile of a typical and middle TPSS in 3 kV DC line. The uncertainty and time-varying features of ERPS are evident in the Figure 13. During the mid-night and early in the morning where the power is low, the passenger trains are not in operation and only freight trains are working. In the proposed system, the centralized configuration is considered where the distributed energy resources and EV fast charging station are implemented close to the TPSS.

In this study, the nominal output power of the photovoltaic system is 1.2 MW, and nominal output power of wind turbines is 3 MW. The maximum output power demand of the EV charging station is 500 KW which usually takes place during peak hours as above mentioned. Likewise, as was disclosed in Figure 13 the power demand of trains at one substation differs from 0.5 MW during off-peak hours to about 6.5 MW during peak hours. The specifications of the various parts of the system are defined in Table 2.

**Table 2.** Smart DC catenary system parameters.

| Item | Description |
|---|---|
| Photovoltaic Generator | 5700 modules 1Soltech 1STH-215-P with 25 series and 228 parallel, 1.2 MW |
| Wind Generator | PMSG, 3 MW, base speed: 9 m/s |
| Storage Unit | 500 V, 4500 Ah Lithium-Ion, response time: 1 s |
| DC Fast-Charging Station (DAB) | 500 kW rating power, 10 kHz switching Frequency |
| TPSS power converter | 10 MVA, 1400 V AC, 3000 V DC, 50 Hz |

Figure 14 shows the daily voltage profile of the DC catenary system that remains on 3000 V dc with +15% and −20% variation. This is an acceptable range for the voltage profile of the DC catenary system since the voltage changes very fast in the electric railway system and the acceptable range is between +20% and −30% [34].

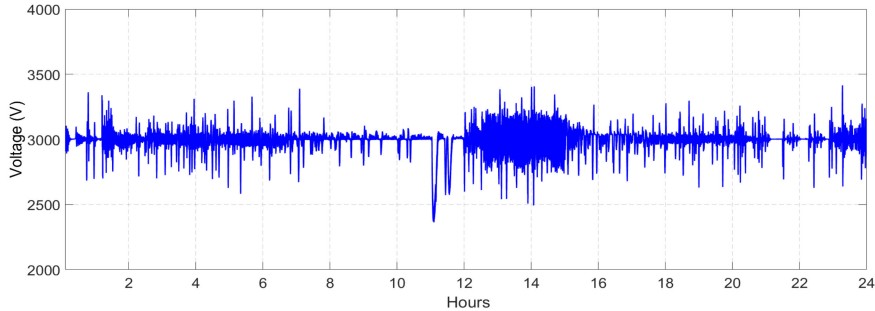

**Figure 14.** Voltage profile of the smart DC catenary system.

In Tables 3 and 4, the filter values and PI coefficients are presented respectively.

Figure 15 discloses the AC side profiles including voltage, current, and power. During normal operation, the utility grid is the major supplier of the smart DC catenary system where the consumed power varies according to the working condition of the system. However, between 11:00 a.m. and 12:00 a.m., there is a blackout in which the utility grid is disconnected.

**Table 3.** Filter design values.

| Item | Filter Value |
|---|---|
| AC side filter values | $L_1 = L_2 = 6.5508 \times 10^{-5}$ H, C $= 2.7067 \times 10^{-4}$ F |
| Storage buck-boost converter | $L_{min} = 5 \times 10^{-5}$ H |
| PV boost converter | $L_{min} = 40 \times 10^{-3}$ H |
| Dual Active Bridge converter | $L_k = 1.65 \times 10^{-4}$ H |

**Table 4.** PI controller parameters.

| Item | PI Coefficients |
| --- | --- |
| AC-DC converter | $K_p = 0.25$, $K_i = 30$ |
| Buck-boost converter | $K_{p-1} = 0.25$, $K_{i-1} = 50$, $K_{p-2} = 0.05$, $K_{i-2} = 10$ |
| Boost converter | $K_p = 0.001$, $K_i = 0.01$ |
| DAB converter | $K_p = 0.085$, $K_i = 0.25$ |
| Wind turbine converter | $K_p = 50$, $K_i = 0.01$ |

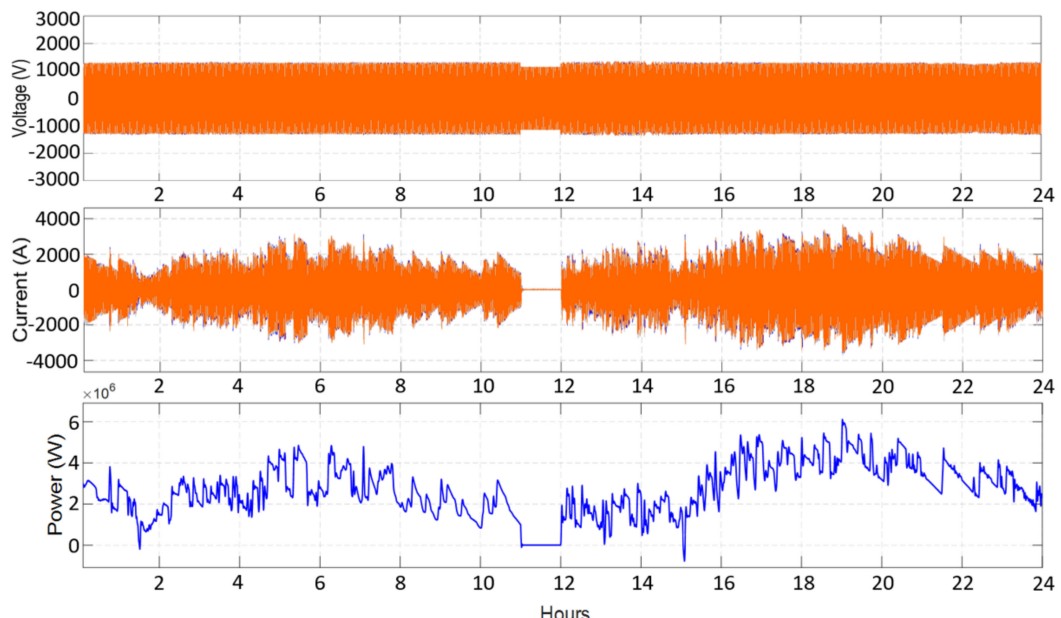

**Figure 15.** AC side voltage, current, and power profile.

The voltage, current, and power generated by PV system is presented in Figure 16. As can be seen, the MPPT algorithm works perfectly and extracts the maximal power according to the radiation and environment temperature.

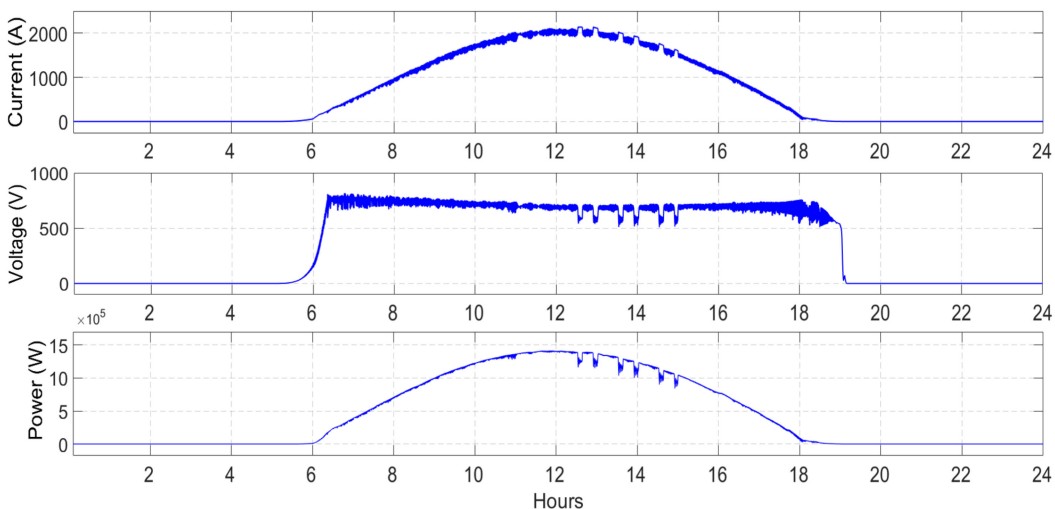

**Figure 16.** PV system voltage, current, and power profile.

In this study, the storage unit acts as a backup in case of a blackout and it is applied due to the intermittent nature of wind and irradiance. The output results of the storage system including voltage, current, and SOC are shown in Figure 17.

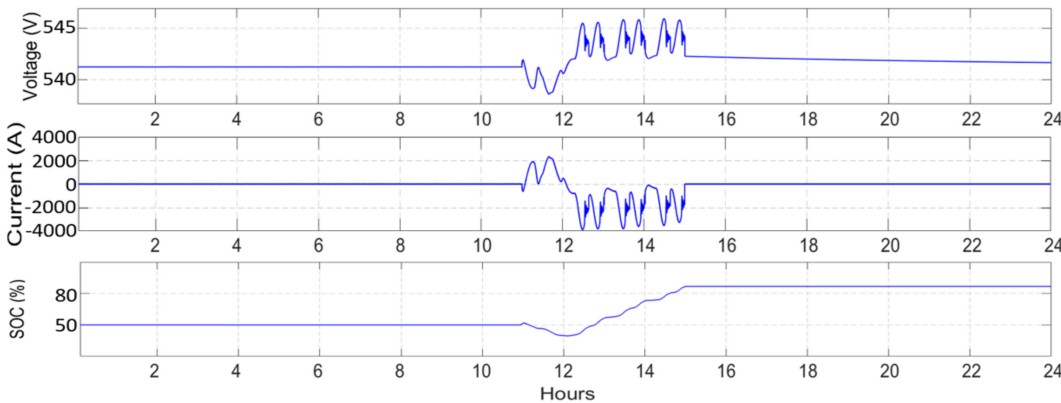

**Figure 17.** Storage system voltage, current, and SOC.

The daily outputs profiles of the wind turbine including voltage, current, and power are presented in Figure 18. As can be noticed, the control structure works properly, and the power profile follows the daily wind speed shown in Figure 11.

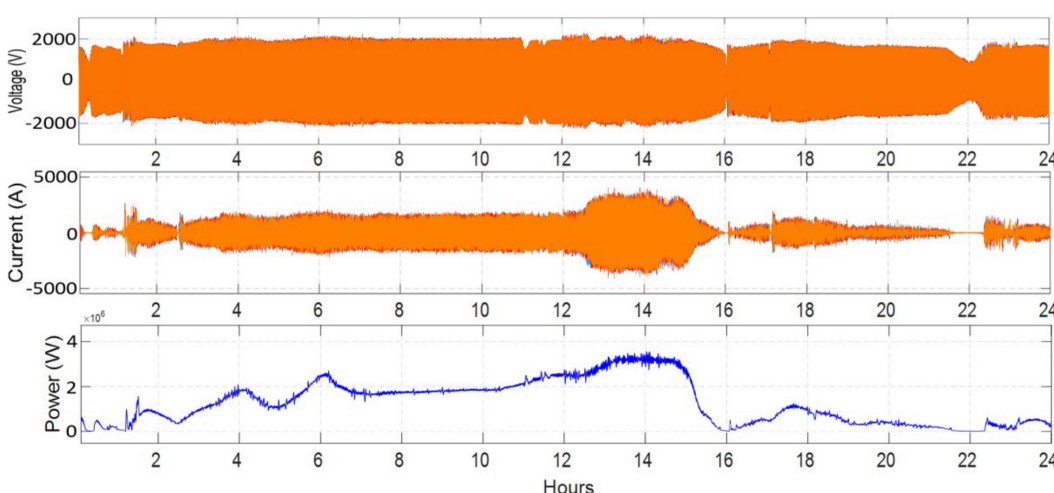

**Figure 18.** The output results of the PMSG wind generator.

Considering the charging infrastructure, the DAB converter is transferring the power from 3 kV dc catenary system to a 500 V DC-link where the charging slots are placed. The output results of charging station including, voltage, current and the power demand are presented in Figure 19. The voltage profile in Figure 19 shows that the DAB converter retains the voltage at 500 V regardless of the number of connected EVs.

Figure 20 represents the output power of various subsystems including utility grid, fast-charging station, renewable energy sources, and the storage unit. As it can be seen, the system is operational in several working modes defined in Section 4. At nights, the ERPS's power demand is at its minimum and both grid and wind generator (if available) are supplying the loads (FM). However, in the mornings, the load's power demand peaks, and thanks to renewables, it can be realized that peak-shaving is happening and drawn power from the grid is much less than the load's power demand (PSM) that is about 30% less. For the rest of the day (around 7:00 a.m. to 8:00 p.m.) the system is mostly working in the feeding mode and the wind/PV/grid system will supply the loads (charging station and trains). Nevertheless, a storage unit is used as a backup, hence, if a blackout takes place (11:00 a.m. to 12:00 a.m.) and the utility grid will be disconnected, the storage system with other renewables will supply the load (EM). In some cases, if the SOC of the storage is 100%, and renewable energy sources generated power exceeds the load's power demand, the extra power will be fed into the main utility grid (RM) that in this model occurs from

3:00 p.m. to 3:05 p.m. The results show that the proposed microgrid based architecture for ERPS can be a promising solution to reinforce the supply line power and also high traffic issues. In other words, by implementing DESs and integrating with existing ERPSs, it is possible to increase the network power capacity and prevent the establishment of more TPSSs or increase the line voltage.

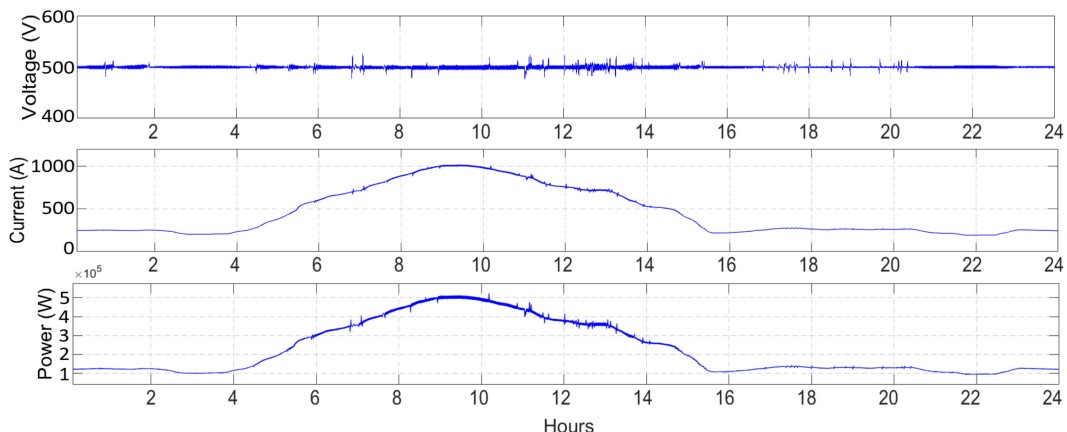

**Figure 19.** The output results of the fast-charging station.

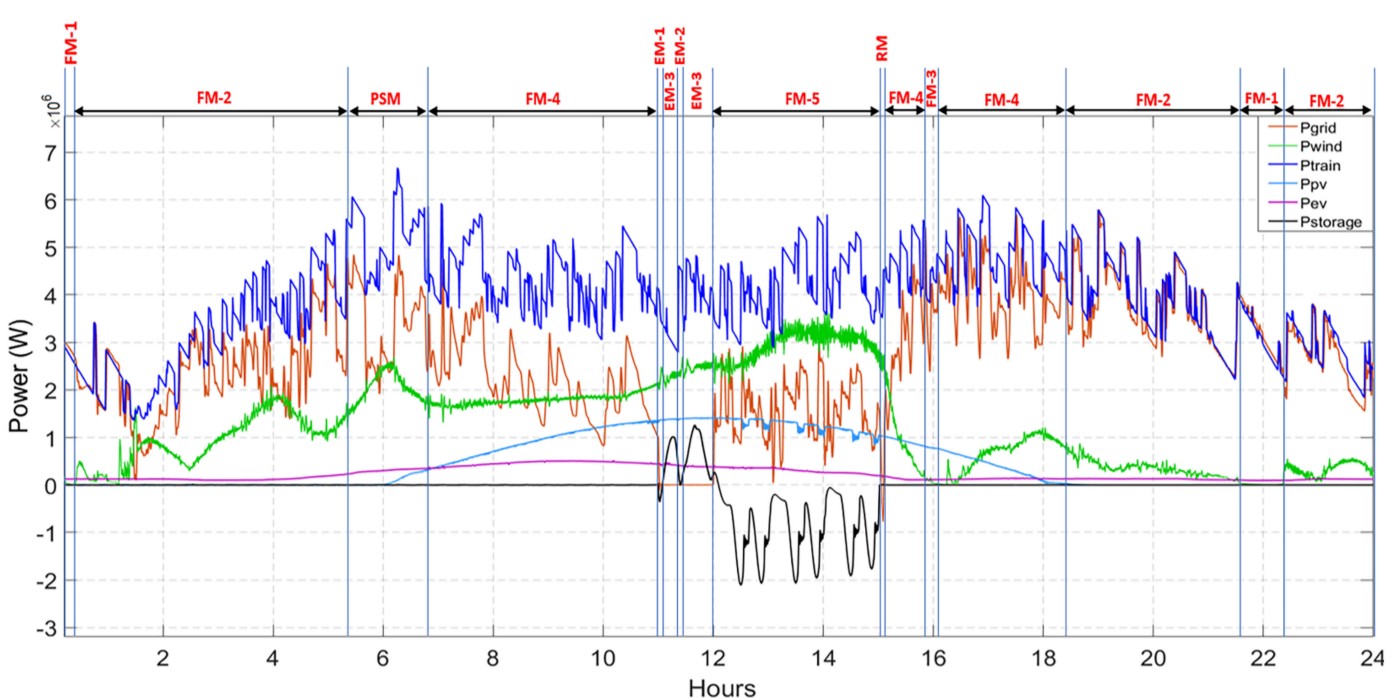

**Figure 20.** Power values for different sources and loads.

## 6. Conclusions

In this paper, a smart DC railway architecture is proposed where two types of renewable energy sources (wind, PV) are integrated into the DC catenary system. Additionally, a storage unit is used to address the intermittency nature of the renewable sources and work as a backup. Furthermore, a DC fast-charging station is integrated into the smart DC catenary system in which the power flow is processed through a bi-directional DC-DC power converter (DAB); it will provide the system with galvanic isolation and two ways power flow. More importantly, the proposed system is evaluated for 24-h by applying the realistic daily input data for radiation, temperature, wind speed, trains power demand profile, and fast charging station power requirements considering power electronic based converters

and their modulation. The feasibility of such a system is validated in MATLAB/Simulink. Worth mentioning, in the proposed architecture, smart DC catenary system is utilized as a DC hub in which the power flow is controlled between power generators and loads. The results confirmed the contribution of such a smart architecture to partial and even temporary full independence from the main utility grid suppliers. Meanwhile, the interlink between two different transportation systems and the capability of power transferring revive the concept of sustainable transportation as the architecture of future power supply systems. In this paper, transferring the DC catenary system of 3 $kV_{dc}$ ERPS into DC hub is presented for accommodating the RESs (wind turbines and PV panels), charging infrastructures, and storage unit in order to increase the capacity of the system and mitigating the voltage drop due to high penetration of loads. Moreover, an PMS is proposed that validates the correct operation of the system under various working modes. However, the next steps can be evaluating the energy management system (EMS), considering the vehicle to grid (V2G) possibility etc. Therefore, to this end, the means of green and low emission transportation will be met by applying the system proposed in this paper.

**Author Contributions:** In preparation of this paper, M.A. and H.J.K. did the methodology; M.A. did the data curation, software-based simulations, and writing—original draft; H.J.K. and M.B. did the validation and visualization; F.C.-D., M.B. and M.S.C. did the resources and funding acquisition; and F.C.-D. and M.B. did the editing and supervision. All authors have read and agreed to the published version of the manuscript.

**Funding:** This research was funded by "Future Unified DC Railway Electrification System" FUNDRES (Shift2Rail) with grant number 881772.

**Conflicts of Interest:** The authors declare no conflict of interest.

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
