# Peer review of "Integration of Distributed Energy Resources and EV Fast-Charging Infrastructure in High-Speed Railway Systems"

_electronics, doi:10.3390/electronics10202555_

Round 1
Reviewer 1 Report
The paper “Integration of Distributed Energy Resources and EV Fast Charging Infrastructure into High-Speed Railway System” proposes a novel smart DC catenary system in which renewable sources, storage systems, and DC fast-charging stations are connected to the overhead DC catenary line of the high-speed railway power system.
The paper addresses an important and interesting topic with a high degree of utility.
The structure of the paper is clear and concise, and the results are conclusive. Some clarification issues have been suggested.
The figures should be first mentioned in text (Figure 1, 9, 10,11 are mentioned after they appear).
It is specified that the proposed model consists of PV modules and WTs as renewable energy production sources.
From the applicability of the implementation of this proposal point of view, it was considered where could wind turbines be installed?
It is specified that WTs have a power of 3MW. How many turbines of what power and dimensions would be used? What dimensions would they have and where could they be mounted? Can they be mounted near the consumption area? Can power and power losses become a problem in terms of the length of the connection cables?
The conclusions section should also contain a series of recommendations.
Author Response
Please see the attachment.
In the attachment, the revised paper is at the beginning and the responses are added to the after.

Reviewer 2 Report
The paper is well written and organized, but the main paper's contribution needs to be highlighted. There are also some questions about some models as follows:
- Would you please state the main contributions of this work to the research community?
- Please, Provide the converter control equations for the PV system shown in fig. 3
- Please, Provide the converter control equations for the ESS system shown in fig. 6
- Why is PMS proposed? Why did the authors not use an energy management system (EMS), mainly since the system contains storage elements such as EVs and ESSs?
- Not all the system parameters are given in the paper, such as The PI controllers’ parameters, the filters’ parameters, etc.. These parameters affect the transient response of the system, especially when using PMS. Can the authors share the Matlab code files with the reviewers to show these parameters and study the system behavior?
- In the simulation, EV is only charging; is V2G proposed in this work?
Author Response
Please see the attachment.
Dear Sir/Madam
Hello, in the attachment, the revised paper with the responses can be found.
Please notice that the first part is the revised paper and the second part is the responses to the reviewers.
Thanks for your time and kindness
Yours sincerely
